# Bending Study of Six Biological Models for Design of High Strength and Tough Structures

**DOI:** 10.3390/biomimetics7040176

**Published:** 2022-10-25

**Authors:** Guangming Chen, Tao Lin, Ce Guo, Lutz Richter, Ning Dai

**Affiliations:** 1College of Mechanical and Electrical Engineering, Nanjing University of Aeronautics and Astronautics, Nanjing 210016, China; 2Large Space Structures GmbH, Hauptstr. 1e, D-85386 Eching, Germany

**Keywords:** biomimetic composites, crack propagation, finite element method, three-point bending test

## Abstract

High strength and tough structures are beneficial to increasing engineering components service span. Nonetheless, improving structure strength and, simultaneously, toughness is difficult, since these two properties are generally mutually exclusive. Biological organisms exhibit both excellent strength and toughness. Using bionic structures from these biological organisms can be solutions for improving these properties of engineering components. To effectively apply biological models to design biomimetic structures, this paper analyses strengthening and toughening mechanisms of six fundamentally biological models obtained from biological organisms. Numerical models of three-point bending test are established to predict crack propagation behaviors of the six biological models. Furthermore, the strength and toughness of six biomimetic composites are experimentally evaluated. It is identified that the helical model possesses the highest toughness and satisfying strength. This work provides more detailed evidence for engineers to designate bionic models to the design of biomimetic composites with high strength and toughness.

## 1. Introduction

Material strength expresses the ability to withstand maximum stress before fracture [1], and material toughness represents the resistance to fracture [2]. High strength and tough materials can prolong product life spans and are, thus, required in many applications [3]. Nonetheless, increasing strength does not always promote its toughness. For instance, to increase strength by increasing stiffness, the toughness can be severely reduced [4]. One practical way to server both strength and toughness is using composite structures [5]. Nevertheless, the mechanical properties of current composite structures are not yet satisfying due to a lack of effective structure models [6]. As a result, it is essential to explore new structure models [7].

Inspired by biological organisms that possess excellent properties of high strength and large toughness, their biological models were adopted in the design of bionic structures [8]. In achieving the biological properties, the structure–mechanics characteristics for increasing strength and toughness were analyzed for several biological models [9]. Beniash et al. [10] found that the dislocation of adjacent crystals in the columnar arrangement of enamel leads to crack deflection, thus, increasing strength and toughness. Suksangpany et al. [11] proposed a theoretical model to elaborate on the toughening effect of crack twisting in the helical model. Liu et al. [12] established quantitative criteria for evaluating the cracking propagations of the sutured interface and revealed the strengthening and toughening effect by the suture interface. Moreover, numerical approaches for predicting mechanical behaviors of biological models were discussed [13]. Gustafsson et al. [14] performed a comprehensive material parameter study by using a 2D extended finite element method (XFEM) interface damage model and simulated crack propagation around an osteon at the microscale. Additionally, biomimetic synthetic structures were fabricated and the mechanical behaviors of several bionic models were experimentally tested. Jia et al. [15] fabricated five biomimetic composites with bioinspired microstructures and characterized their crack resistance features. Agarwal et al. [16] fabricated a biomimetic tough helicoidal structure inspired by mantis shrimp’s dactyl club and experimentally investigated its mechanical properties; these works demonstrated that high strength and toughness can be combined in bionic structures [17].

Although the above research provides significant knowledge in designing bionic structures, bionic composites with high strength and toughness are not always successful. This is largely attributed to the mechanisms for obtaining high strength and toughness of biological models that are not clearly understood [18]. In addition, the numerical illustrations of the strengthening and toughening mechanisms of the biomimetic composites are rarely performed [19]. In addition, experimental verifications of the mechanisms of biological models are not completely investigated [20]. To effectively design bionic structures with high strength and toughness, it is essential to investigate the strengthening and toughening mechanisms of the fundamentally biological models through comprehensive studies, including both numerical and experimental tests [21,22]. 

This paper provides analysis on the mechanisms of six fundamental biological models using numerical and experimental approaches. In Section 2, the six biological models that can have high strength and toughness are extracted from biological organisms; Section 3 illustrates the numerical approaches for predicting their strengthening and toughening mechanisms. In Section 4, the experimental tests of the bionic structures are presented. Finally, conclusions are presented in Section 5.

## 2. Biological Models

Biological organisms of high strength and toughness have varied structures and function mechanisms with respect to different size scales [13]. From micro- and meso- scales, this section summarizes six fundamental structure models, namely, layered, columnar, tubular, sutured, helical and sandwich. The corresponding strengthening and toughening mechanisms are also explained. These six models are considered to comprise most common structure features for high strength and toughness in biological organisms. These models can also be integrated for the structure design and applications.

### 2.1. Layered

The layered model in biological organisms is composed of multiple overlapped stiff platelet material and soft viscoelastic matrix (Figure 1) [23]. Layered structures appear in the microstructure of many biological organisms, such as nacre shells (Figure 2a) [24], insect exoskeletons (Figure 2b) [25], fish scales (Figure 2c) [26] and deep-sea sponges (Figure 2d) [27]. For the mechanistic study of this model, Gu et al. [28] presented a systematic investigation to elucidate the effects of the volume fraction of stiff platelet materials on the mechanical response of nacre-inspired additive manufactured layered composites. Experimental and finite element simulation of tensile tests on single edge-notched samples were carried out. Results reveal the importance of the stiff phase in carrying the load and the soft phase in transferring the load through the platelet by shearing. Narducci et al. [29] designed and tested a bio-inspired carbon fibre/epoxy composite with a layered microstructure. Analytical models were developed to predict the energy dissipation and crack deflection properties. In addition, three-point bend tests are carried out to observe the fracture behaviors. The results showed that the layered structure is capable of deflecting the crack, avoiding sudden failure in the most highly loaded cross section of the specimen. Liu et al. [30] investigated the interfacial strength and fracture mechanism of the ‘brick-mortar’ structure using micro-sized cantilever beam and bend samples. The crack propagation path was also investigated via experiment and finite element modelling. The results were compared with the fracture mechanics. It confirmed that crack deflection to the aragonite/biopolymer interface contributed to a high overall toughness. 

These pieces of research revealed the strengthening and toughing mechanism of the layered model via experiments, simulations and numerical analysis. When the platelets are subjected to tensile loads, the matrix transfers the loads by shearing interfaces, so that the force for pulling out platelets increases, and thus enhancing strength and toughness of the model [31]. In addition, crack deflection and crack bridging at the interfaces also improve toughness. 

### 2.2. Columnar

The columnar model consists of stiff columns wrapped in a soft matrix, as is shown in Figure 3. In this model, the parallel columns can be placed at intervals or compacted. This columnar structure is often found in mineralized organisms, such as in tooth enamel (Figure 4a) [32]. They are also widely observed in non-mineralized soft biological materials, such as bamboo fibers (Figure 4b) [33], silk (Figure 4c) [34] and spider silk (Figure 4d) [35]. To assess the strength and toughness of this model, Bajaja et al. [32] quantitatively studied the crack growth resistance and fracture toughness of the columnar structure of human tooth enamel by three-point bending tests. The results revealed that bridging the organic matrix promoted crack closure. In addition, microcracking due to loosening of columns can lead to energy dissipation. Yeom et al. [36] fabricated enamel-inspired columnar nanocomposites, consisting of columns with polymeric matrix. The Young’s modulus and hardness of the samples were experimentally and numerically measured by performing nano-indentation tests. The results confirmed that high performances of the columnar structure were attributed to efficient energy dissipation in the interfacial portion between the columns and the organic matrix.

As the researchers demonstrated above, the high viscoelasticity between the columns and the matrix improves tensile strength [37]. When the column fractures are being pulled, the slip between column and matrix and the microcrack deflection at the matrix–column interface contribute to energy absorption and dissipation, thus, enhancing structure toughness.

### 2.3. Tubular

The tubular model refers to parallel stiff tubes staggered in soft matrix, as is shown in Figure 5. These structural elements are commonly found in biological materials that resist impact and compression such as horse hooves (Figure 6a) [38], ram horns (Figure 6b) [39], dentin (Figure 6c) [40] and whale baleen (Figure 6d) [41]. To reveal the mechanisms of this model, Giner et al. [42] studied the elasticity and toughness properties by comparing experimental tests with finite element simulations. In particular, three-point bending tests have been performed and the growth of cracks were simulated through 2D XFEM based on a damage model of maximum principal strain criterion. The simulation revealed that cracks were frequently arrested or deflected when they encountered a tubule, which agreed with the experimental results. Wang et al. [41] fabricated a tubular structure consisting of non-mineralized filament matrix and tubules composed of mineralized lamellae by using 3D printing. Three-point bending tests were conducted both on the transverse and longitudinal orientations of the specimens to obtain the J-integral of the structure toughness. The results revealed that the cracks growing in the transverse direction were arrested and redirected along the tubules. As a result, the resistance to fracture was enhanced and the J-integral was increased.

The research above inferred the behaviors of the tubulars for the strengthening and toughening mechanisms. The stiff tubes can resist highly compressive external forces. The organized pores of tubes absorb compression energy and enables crack deflection as fracture occurs [43]. Moreover, they can arrest crack growth through removing the stress singularity at the crack tip or by collapsing the tubules when compressed to improve fracture toughness [44]. 

### 2.4. Helical

The helical model in biological organisms can be described as successive fiber layers in a weak matrix (Figure 7) [45]. The fibers in each layer have relative rotational angles (*Δθ*) and can also have angle deflection in vertical directions [46]. A common type of the helical structure is a periodically assembled uniaxial fiber layers, which is referred to as Bouligand structure [47]. Helical model is commonly seen in mantis shrimp dactyl club (Figure 8a) [48], fish scales (Figure 8b) [49], beetle exoskeletons (Figure 8c) [50], deep-sea sponges (Figure 8d) [51] and other similar organisms. To evaluate the strengthening and toughening mechanisms of this model, Suksangpanya et al. [46] carried out a theoretical and experimental combined approach to estimate the structure toughness using a 3D printed biomimetic composite material. Three-point bending tests were conducted and the effects of structural parameters were mathematically modeled. It was found that crack twisting driven by the fiber architecture, crack branching and delamination, were the main reasons for avoiding catastrophic failure. Yin et al. [49] adopted a fracture model of an anisotropic phase-field to predict the toughness of Bouligand structures. Moreover, the Bouligand structure was fabricated and tension tests were performed. The results revealed the advantages of Bouligand structures in promoting the isotropy and enhanced fracture toughness properties.

The above research indicated that the helical arrangement can result in the spatial variation of the driving force, which provides significant resistance to multi-directional loads [52]. Moreover, the dislocation of the fibrous layers decomposes the cracks and enables them to deflect, twist or bifurcate, which are responsible to the increase in structure toughness [53]. 

### 2.5. Sutured

The sutured model contains stiff suture teeth and a compliant soft interface layer, generally possessing wavy or interdigitating interfaces (Figure 9). The sutured model usually appears in the interfaces of biological organisms, where it is necessary to adjust intrinsic strength and flexibility, for instance, deer’s skulls (Figure 10a) [54], boxfish scales (Figure 10b) [55], turtle shell (Figure 10c) [56] and the pelvis of three spine sticklebacks (Figure 10d) [57]. To understand the mechanisms for the high strength and toughness of the model, Cao et al. [57] carried out numerical and experimental study based on 3D printed bionic suture joint specimens. The tensile failure behavior of specimens was systematically studied and the failure mechanisms of the joints were explored by studying the influences of critical geometric parameters. It revealed that the tooth-shaped or sinusoidal curve of the suture interface can improve the strength and toughness of the structure. Rivera et al. [58] fabricated a series of biomimetic composites with sutured structure, including ellipsoidal geometry and laminated microstructure. Tensile tests and 2D finite element simulations were conducted. The results revealed that the suture structure with ellipsoidal geometry can provide mechanical interlocking, which increases strength and toughness significantly.

The above studies implied that the sutured interfaces can transfer and distribute loads, thus, reducing concentrating stresses and increasing structure strength [59]. The interlocking of two stiff components occurs at the interface under tensile load, which can improve strength and toughness [12]. Furthermore, suture tooth fracture and interfacial shear failure dissipate energy. 

### 2.6. Sandwich

The sandwich structure refers to inner cellular structure wrapped by dense outer shell. The inner cellular structure can be connected or disconnected, having two- or three-dimensional periodic cores or foam and one such is three-periodic minimal surface (TPMS) structure (Figure 11) [60,61]. Sandwich structures are commonly found in stiff but lightweight biological organisms; typical examples are toucan beaks (Figure 12a) [62], skeletons (Figure 12b) [51], antlers (Figure 12c) [63], horseshoe crab exoskeletons (Figure 12d) [64]. To show the strengthening and toughening mechanisms of this model, Bang et al. [65] conducted experimental and 3D simulation studies on the compression property of sandwich based on the compression tests. The simulation results were consistent with the experiments, confirming that the porous sandwich structure has excellent energy absorption capability. Pathipaka et al. [66] fabricated honeycomb and foam sandwich structures and demonstrated the excellent energy absorption capabilities due to their porous characteristics.

As the researchers illustrated above, the dense shell of the sandwich structure is rigid, which contributes to high strength whilst the inner core or foam effectively absorbs energy under bending and compression forces. In particular, the inner porous material deflects cracks by forcing them to pass through pore or foam surfaces as failure occurs, which enhances structure toughness [67]. 

## 3. Numerical Modelling

Despite many studies having experimentally demonstrated the high strength and toughness of the above six bionic models, the simulations of crack propagations using 3D models are also yet to be conducted. The numerical modelling of mechanical behaviors can reveal strengthening and toughening mechanisms of the bionic models and can, thus, improve structure design. The XFEM in Abaqus [68,69] can model the crack propagation of 3D composite structures. To demonstrate strengthening and toughening mechanisms, this section numerically illustrates the mechanical behaviors of the six bionic models.

### 3.1. Simulation Setup

The analytical modelling of the three-point bending test of the six bionic models can be used to predict crack propagation behaviors [46]. The six bionic models are composed of soft matrix and stiff material. For layered, columnar, tubular, sutured and helical models, the soft and stiff parts are individually imported to Abaqus for assigning different materials. The ‘Retain’ function is applied to merge these two materials into a single part. For the sandwich model with distinguished boundary from soft to stiff phases, its materials are conveniently applied after the whole single part is imported. The support and loading pins are both structural steels. The simulation parameters for soft and stiff materials are referred to in the experimental samples that are given in Table 1. 

Crack initiation adopts the principle that local stress exceeds the material maximum principal stress [68], which can be expressed as:(1)$$f=\left\{\frac{\langle \sigma_{max}\rangle }{\sigma_{max}^{0}}\right\}$$
in which $\sigma_{max}^{0}$ is the allowable maximum stress. The symbol “⟨⟩” represents the Macaulay bracket with the usual interpretation (i.e., $<\sigma_{max}$ ≥ 0 if $\sigma_{max}$ < 0 and <$\sigma_{max}$ ≥ $\sigma_{max}$ if $\sigma_{max}$ ≥ 0).When the maximum stress of the material $\sigma_{max}$ is greater than the threshold $\sigma_{max}^{0}$, crack initiates in the modeled 3D structure. The stresses (both normal and shear) are in three directions, which give three normal stresses and three shear stresses. The maximum principal stress $\sigma_{max}=\mathrm{Max}\left(\sigma^{\prime },\;\sigma^{\prime \prime },\;\sigma^{\prime \prime \prime }\right),\mathrm{where},\;\sigma^{\prime },\;\sigma^{\prime \prime }\mathrm{and}\;\sigma^{\prime \prime \prime }\;\mathrm{are}$ the three normal stresses at their own orientations. These three principal stresses can be obtained by solving the following cubic equation, [70]
(2)$$\sigma^{3}-I_{1}\sigma^{2}+I_{2}\sigma -I_{3}=0$$
where
(3)$$\left\{\begin{array}{c} I_{1}=\sigma_{x}+\sigma_{y}+\sigma_{z}\\ I_{2}=\sigma_{x}\sigma_{y}+\sigma_{y}\sigma_{z}+\sigma_{z}\sigma_{x}-\tau_{xy}^{2}-\tau_{yz}^{2}-\tau_{zx}^{2}\\ I_{3}=\sigma_{x}\sigma_{y}\sigma_{z}+2\tau_{xy}\tau_{yz}\tau_{zx}+\sigma_{x}\tau_{yz}^{2}-\sigma_{y}\tau_{Zx}^{2}-\sigma_{z}\tau_{xy}^{2} \end{array}\right.$$

Equation (2) gives three roots of the three principal stresses for the given three normal stresses ($\sigma_{x}$, $\sigma_{y}$ and $\sigma_{z}$), which are:(4)$$\left\{\begin{array}{c} \sigma^{\prime }=\frac{I_{1}}{3}+Rcos\frac{\varphi }{3}\\ \sigma^{\prime \prime }=\frac{I_{1}}{3}+Rcos\frac{\varphi +2\pi }{3}\\ \sigma^{\prime \prime \prime }=\frac{I_{1}}{3}+Rcos\frac{\varphi +4\pi }{3} \end{array}\right.$$
in which:(5)$$\left\{\begin{array}{c} R=\frac{2}{3}\sqrt{I_{1}{}^{2}-3I_{2}}\\ cos\varphi =\frac{2I_{1}^{3}-9I_{1}I_{2}+27I_{3}}{2{\left(I_{1}{}^{2}-3I_{2}\right)}^{\frac{3}{2}}} \end{array}\right.$$

Referring to the soft material used in experiments (Table 1), the maximum principal stress for soft matrix is set at 6.5MPa. For stiff materials, a range of 350–550 Mpa is applied in simulation models considering the variation of mechanical properties of experimental samples. In addition, it is necessary to designate the displacements at failure to represent the total displacements triggering material damages. A small value of this parameter means that the material is brittle, which is easier to damage, compared to soft material under the same load. According to the elongations for the used materials in this work, the displacement at failure of soft matrix is set at 0.2 mm, while it is 0.01 mm for stiff material. The viscosity coefficient can be determined at 0.005, according to the properties of the two materials. These parameters and values are also listed in Table 2.

Figure 13 shows the simulation setup for the six bionic models after meshing in Abaqus. The bionic model is symmetrically tied to two bottom supporters of semicircular columns. The materials for supporters, soft and stiff materials, are described as grey, white and yellow. Pre-crack is utilized for initiating initial propagation. The pre-crack is set to middle bottom of each bionic model. A specified displacement is applied to the top semicircular column, which enables subsequent crack propagations.

### 3.2. Simulation Results

Figure 14 presents the simulations of crack propagations by modelling the three-point bending tests of the six bionic models. The color scale denotes the deformations during propagations. The left column displays the results of the whole models at the final step, and the right column zooms in the deformation or fracture features of the single material. For the layered model of Figure 14a, it shows that crack has extended straightly to the stiff platelet. However, it does not affect the crack of the stiff platelet, as predicted by other researchers [71]. This means that the maximum stress of the stiff material is lower than the threshold. Figure 14b shows that the crack propagates a little further in the soft matrix after it has approached the column. It inferred that the solid stiff material can resist higher stress, and thus is effective for resisting crack propagation. By contrast, Figure 14c resembles that crack extends into stiff part, indicating that the maximum tensile stress is higher than the fracture strength of the stiff part. As the tubular factures, the crack of the whole model can be restrained at stiff phases, which presents the failure of whole model. Figure 14d demonstrates that crack defections occur in the soft matrix due to the helical arrangement of helical columns. The distortions of the stiff fibers during crack propagations are observed. Figure 14e shows that crack propagation defects as it extends to the soft phase. This is beneficial to absorbing kinetic energy and also restraining cracks within the soft phase. Figure 14f depicts slight crack growth in the minimum surface structure. This is because the minimum surface structure of the porous core can enable large deformation to disperse the concentrated stress, which alleviates stress transition and absorbs energy.

To sum up, this section presents numerical investigations on the fracture behaviors of the six biological models using XFEM in Abaqus. By analyzing crack propagation behaviors in the simulations of three-point bending tests, the strengthening and toughening mechanisms of the six bionic models are further revealed. The results are consistent with the analysis in the previous section.

## 4. Experimental Verifications

To enhance application of bionic structures to engineering components, six biomimetic structures based on the six bionic models are fabricated. Experimental three-point bending tests are conducted to the fracture behaviors of crack propagations. In addition, the strength and toughness of the six bionic structures are evaluated.

### 4.1. Sample Fabrications

For the six samples, the soft and stiff materials are, respectively, Ajilus30 and hard resin VeroBlackPlus, both produced by the company of Stratasys. The 3D printing technology is PolyJetT, and the provider of the samples is DEE 3D. The size errors between 3D models and fabricated samples are between 0.1–0.2 mm. The deviations for the mechanical properties shown, such as Young’s modulus and tensile strength, are around 18%. These deviation have not influenced the results according to simulations for the tested parameter ranges. Referring to standard test model by ASTM [72], the sample sizes for three-point bending test are illustrated in Figure 15. The length, width and height are, respectively, 135 mm, 15 mm and 30 mm for all specimens. The initial cracks are applied for enhancing crack extensions. The notch sizes are 5.55 mm in height and 1.8 mm in width and have a crack closing angle of 60°. The bionic specimen designed from these six bionic models are shown in Figure 16. Using 3D printing technology, these six bionic structures were fabricated, as presented in Figure 17. The materials properties for the soft and hard parts of each specimen are listed in Table 1. The mass of layered, columnar, tubular, sutured and helical samples are around 70.5 g, and the sandwich sample is 30.1 g.

### 4.2. Experimental Tests

Referring to three-point bending test standard of ASTM [72], the experimental tests of the fabricated six bionic structures are carried out. Figure 18 illustrates three-point bending test of the bionic layered structure using a microcomputer controlled electronic universal testing machine manufactured by WANCE Ltd. (Shenzhen, China). The support span of the setup is 100 mm. The loading rate maintains 10.0 mm/min for whole test period, and the reaction forces by the specimen are recorded. 

Figure 19a–f presents test results of the six bionic structures. For the layered model, the crack propagated through soft phases for whole failure process and deflected when approaching stiff materials. This indicated that the stiff bricks did not support the high compression load. For the columnar model, the volume fraction of this model is slightly lower than all others, which causes the larger deformation of the soft matrix. Despite the large deformation bending of the specimen, the crack of model hardly propagated and the specimen remained intact. For tubular model, the stiff tubulars suffered to brittle fractures and the crack propagates in a straight line. For the helical model, the crack twists along layers of the alignment fibers. In addition, delamination occurs as the crack extends to different layers. For the suture model, initially the pre-crack propagated a small distance. After this, the whole structure experienced brittle fracture, which generated a new crack path. This is due to the fact that the maximum stress of the stiff material is higher than the threshold, and this model contains a larger part of the stiff material. For the sandwich model, the crack expanded slowly in soft materials and the spaceman fracture to failure when it reaches the outer hard shell. It is concluded that most of the crack propagation features were predicted by the simulations. All models suffer from failure except the column structure, which may indicate that the bionic column structure exhibits high toughness. 

To claim the actual improvement of nature-inspired designs of the six bionic structures, a reference model based on conventional laminated composites was used. [73]. Laminated composites are usually comprised of layers with different materials, which are widely used in industry due to remarkable strength and toughness [74]. Figure 20a shows the laminated structure of the reference model having the same sizes as shown in Figure 15. It consists of five layers of 1.5 mm thick stiff material, which are arranged in parallel in soft matrix. In this reference 3D structure model, the white laminated layers are the stiff material, which have the same mechanical properties as the black material in previous models. Figure 20b shows the bending test of the reference model using the same machine (WANCE, Shenzhen, China), and Figure 20c presents the test result of this sample. Due to the relatively high fraction for the soft matrix, crack propagation was not observed either. 

The force–displacement curves of the six structures and the reference model are plotted in Figure 21. Initially, the compression forces increase with crack growth. Excluding the suture model, the forces for models maintain a decreasing trend after reaching a peak. Specifically, the layered and helical models decrease in a wave form, which is due to the fact that the arrangement of stiff parts affect crack deflections. The tubular experiences a zigzag shape, which is attributed to the individual fracture of the stiff tubes. The column model did not suffer from dramatic drop, as its crack extension was restricted by stiff columns. The first drop happened to the suture is because the crack extended to soft part. Eventually, both suture and sandwich models suffer sharp drop because brittle failure occurs as cracks extend in the stiff phases. The reference model did not experience crack expansion due to the large volume fraction of soft materials.

Due to the seven structures being the same size, their strength can be assessed by the maximum compression force during the three-point bending tests. The toughness of the six bionic structures can be quantified using J-integral of nonlinear fracture [75,76]:
(6)$$J=\frac{2}{Bb}\int_{0}^{S}FdS$$
where *B* and *b* are, respectively, the thickness of the specimen and the remaining ligament of the specimen, *B* = 15 mm, *b* = 5.55 mm; *F* is the applied load; *S* is the loading displacement. The results of the maximum compression forces and toughness of the six bionic structures are presented in Figure 22.

Noting that the layered, columnar, tubular, sutured and helical samples have equivalent mass (70.5 g), it is observed from Figure 22 that, compared to the reference model, the tubular, helical and suture structures have a relatively greater strength and the layered, columnar and helical structures have a relatively greater toughness. The layered model exhibits the lowest strength due to the fact that the stiff bricks could not sustain high loads. The sutured model exhibits the highest strength. Nonetheless, its toughness is the least among the five models. This is because the fraction of the stiff material is too high which caused brittle failure. The helical model may be the best model, since it exhibits the highest toughness and relative high strength. For sandwich model, which had the lowest strength and toughness, its material mass is less than half of the other models. 

## 5. Discussions

This work focuses on investigations of how structure elements affect the strength and toughness of bionic models, which are inspired by nature. To extract biological models with properties of high strength and toughness, biological function mechanisms of structure elements in biological organisms are investigated. Six biological models are extracted based on a broad range of biological organisms over micro- and meso-scales. The presented analysis of biological mechanisms is from a biological systems perspective, and it is hypothesized that the six bionic models are of significant potential in achieving high strength and toughness, owing to intrinsic composite structures [13,77]. By providing detailed analysis of the strengthening and toughening mechanisms for the structure elements in biological models, the effectiveness of using the bionic models for increasing strength and toughness are seen. 

By applying analytical models for assessing material failure, numerical estimations of strength and toughness of the bionic models are conducted. In the work of assessing material failure [78], the researchers successfully modelled stochastic fracture behaviors of ceramics, with respect to different microstructural features. The finite element analysis methodology adopted a fracture mechanics mode to predict the strength scatter in ceramics. Nonetheless, the models are simply applicable to a single material. The applicability to composite materials has not been verified. By contrast, in our work, the fractures over different materials can be modelled. For the work in numerical estimations of strength and toughness [79], the authors developed a constitutive model and applied to finite element simulations. Because the damage process was formulated based on the fracture mechanics of an isotropic damage model, the fracture behaviors of 2D composite structure are reasonably predicted. However, their damage model has not been tested for predicting 3D composite structures. On the contrary, our work demonstrated the simulations of crack propagation for 3D composite structures. The simulations also provide details on how the structure elements gain high strength and toughness by changing crack trajectories and alleviating concentrated stress. For a more realistic resemblance of the experiments, the contact parameters will be improved.

Using prevailing 3D printing technology, the biomimetic composites, based on the six bionic models, can be conveniently fabricated. Experimental tests confirm the validity of the hypothesis that structure elements can change crack propagation directions and release stress by mechanical deformations [15]. The higher fraction of the soft matrix causes more deformations but lower strength for the layered, columnar, tubular, sutured and helical model. In addition, quantitative comparisons on the strength and toughness of the six biomimetic composites are compared, which can be referred to when applying bionic models for the design of high strength and tough structures [80]. For a combination of these structural elements, which can have additional effect on mechanisms, future investigations will be undertaken. To better illustrate the best model in different conditions, new investigations for evaluating mechanical properties, such as a tensile, torsion of the six models shall also be conducted. This is because at the end for both civil and tissue engineering, the structural choice will rely on this type of complete investigation.

## 6. Conclusions

To promote bionic design of high strength and tough structures, this paper classified six basic biological models from micro- and meso-scales of biological organisms. For the numerical analysis for explicitly illustrating strengthening and toughening mechanisms, six biological models were conducted. Most features of the crack propagation behaviors of the six specimens were successfully predicted. The strength and toughness of six bionic structures were assessed using three-point bending tests. It demonstrates that the arrangement of soft and stiff parts affects crack propagation behaviors, and thus dominates the strength and toughness. The solid columns and porous core can resist higher stress, compared to hollow tubulars for the test sizes. For the layered, columnar, tubular, sutured and helical models, a higher fraction of the soft matrix causes more deformations but lower strength, and higher fraction of the stiff material can cause brittle failure and lower toughness. The experimental tests showed that the helical exhibit highest toughness and also high strength. Although the sandwich model shows the lowest strength and toughness, its material cost is much less. This work provides straightforward basis for engineers to designate bionic models and applies to the design process of biomimetic structures with excellent mechanical properties.

## Figures and Tables

**Figure 1 biomimetics-07-00176-f001:**
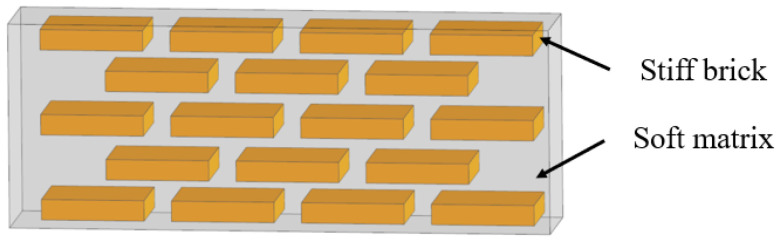
Schematic representation of the layered model.

**Figure 2 biomimetics-07-00176-f002:**
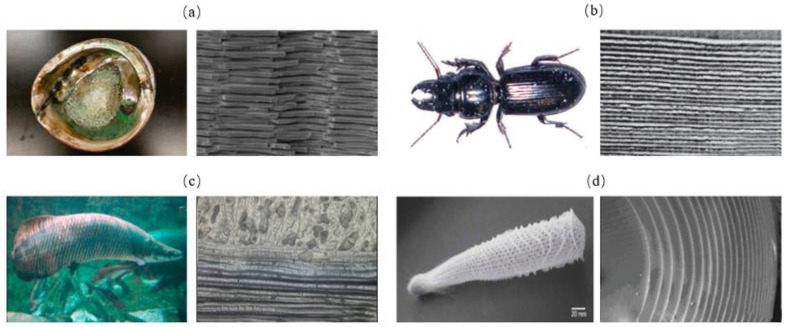
Layered model in biological organisms. (**a**) nacre shell [24] Copyright©2009, Progress in Materials Science (**b**) beetle shell [25] Copyright©2008, Advanced Engineering Materials (**c**) fish scale [26] Copyright©2012, Journal of Materials Research (**d**) concentric layer of deep-sea sponge [27] Copyright©2008, Journal of Materials Research.

**Figure 3 biomimetics-07-00176-f003:**
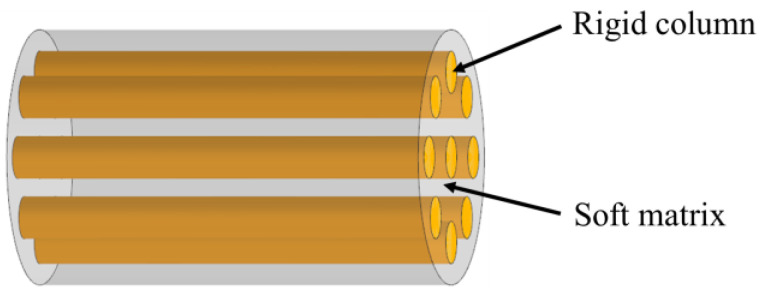
Schematic diagram of the columnar model.

**Figure 4 biomimetics-07-00176-f004:**
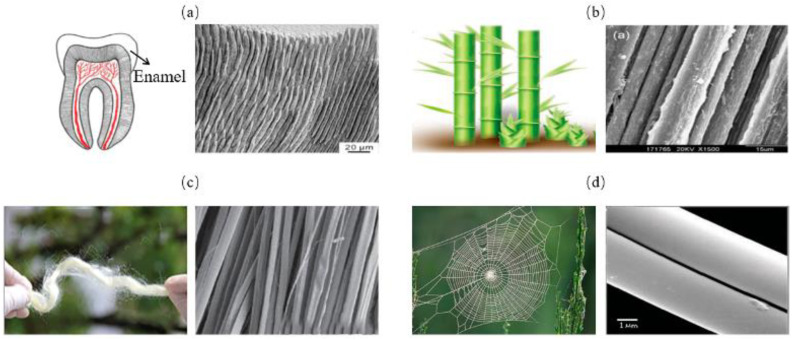
Columnar model in biological organisms. (**a**) tooth enamel [32] Copyright©2009, Biomaterials (**b**) bamboo [33] Copyright© 2020, Advanced Materials (**c**) silk [34] Copyright©2018, ACS Nano (**d**) spider silk [35] Copyright©2020, Advanced Materials.

**Figure 5 biomimetics-07-00176-f005:**
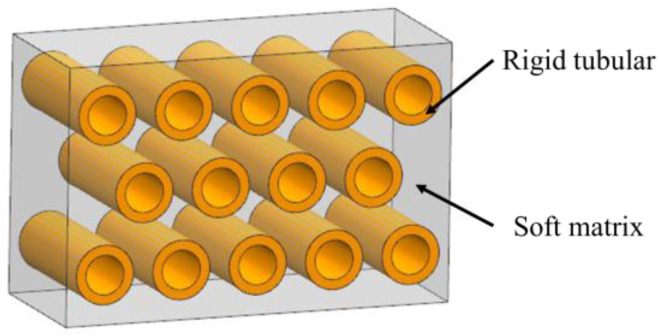
Schematic diagram of the tubular model.

**Figure 6 biomimetics-07-00176-f006:**
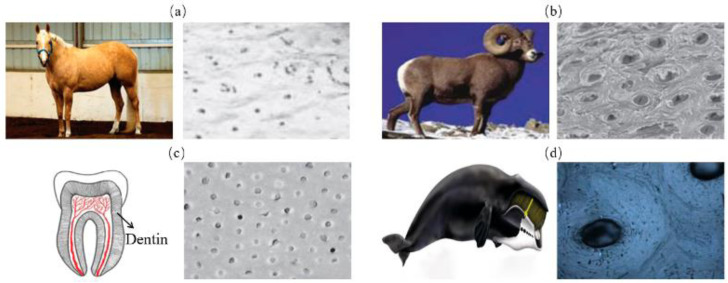
Tubular model in biological organisms. (**a**) horse hooves [38] Copyright©1997, Journal of Experimental Biology (**b**) ram’s horn [39] Copyright© 2010, Acta Biomaterialia (**c**) dentin [40] Copyright©2012, Non-Metallic Biomaterials for Tooth Repair and Replacement (**d**) whale’s baleen [41] Copyright©2019, Advanced Materials.

**Figure 7 biomimetics-07-00176-f007:**
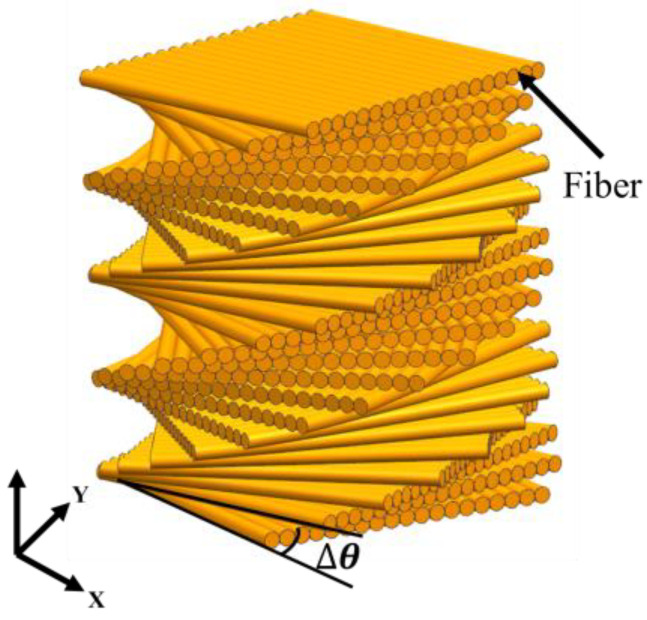
Schematic diagram of the helical model.

**Figure 8 biomimetics-07-00176-f008:**
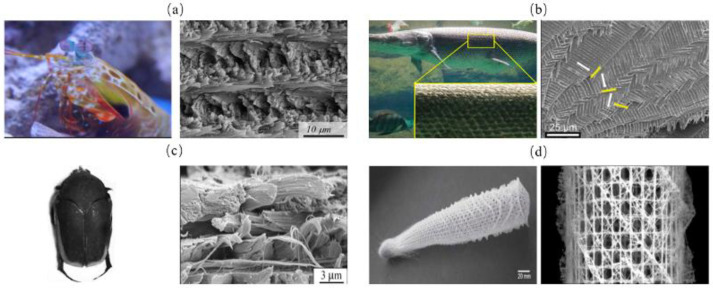
Helical model in of biological organisms. (**a**) mantis shrimp appendage [48] Copyright©2012, Science (**b**) fish scale [49] Copyright©2019, Journal of the Mechanics and Physics of Solids (**c**) beetle exoskeleton [50] Copyright©2019, Acta Biomaterialia (**d**) deep sea sponge [51] Copyright©2007, Progress in Materials Science.

**Figure 9 biomimetics-07-00176-f009:**
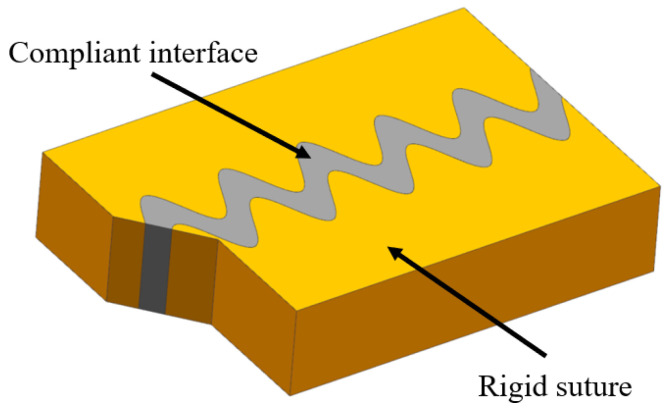
Schematic diagram of the sutured model.

**Figure 10 biomimetics-07-00176-f010:**
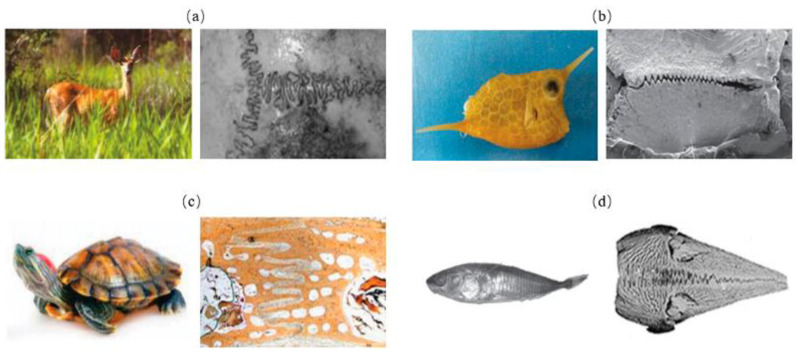
Biological tissue sutured structures. (**a**) deer skull [54] Copyright© 2009, Journal of Morphology (**b**) boxfish scales [55] Copyright©2015, Acta Biomaterialia (**c**) turtle shell [56] Copyright©2009, Advanced Materials (**d**) spiny fish pelvic bone [57]. Copyright©2019, Journal of the Mechanical Behavior of Biomedical Materials.

**Figure 11 biomimetics-07-00176-f011:**
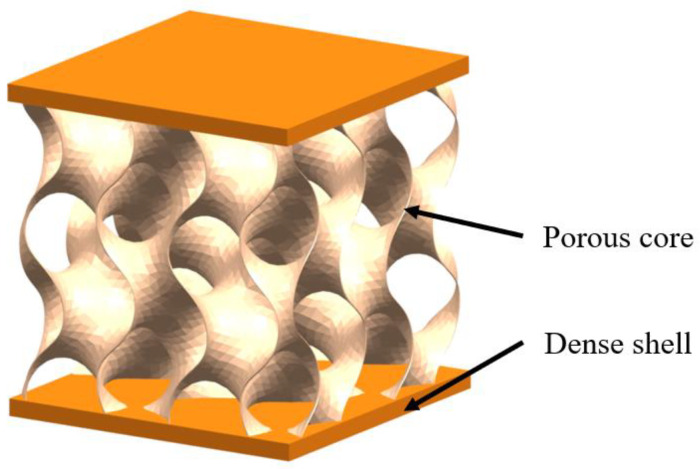
Schematic diagram of the sandwich model with minimal surface structure core.

**Figure 12 biomimetics-07-00176-f012:**
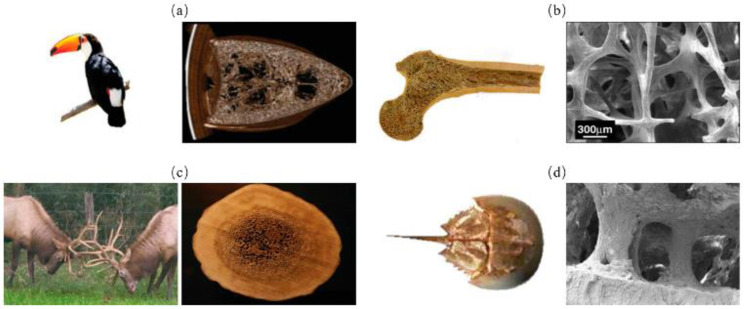
Sandwich model in biological organisms. (**a**) toucan beak [62] Copyright©2005, Acta Materialia (**b**) human trabecular bone [51] Copyright©2007, Progress in Materials Science (**c**) antler [63] Copyright©, 2010Acta Biomaterialia. (**d**) horseshoe crab [64]. Copyright©2018, Progress in Materials Science.

**Figure 13 biomimetics-07-00176-f013:**
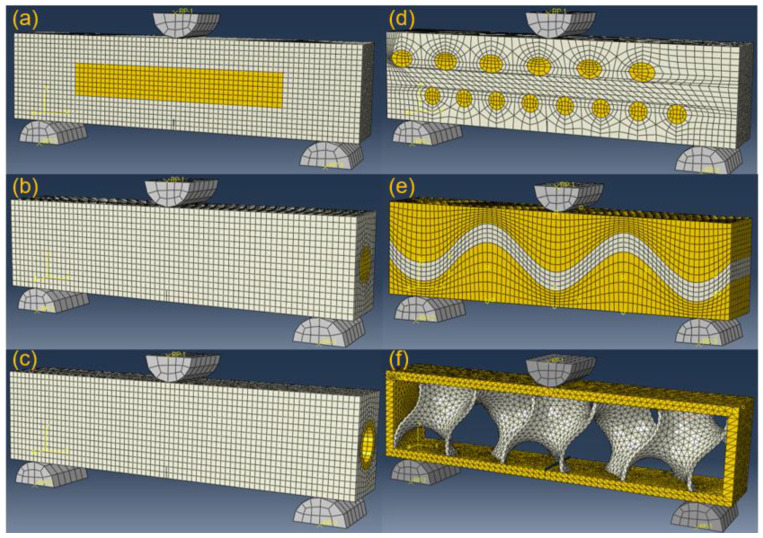
Simulation setup of three-point bending test of the six bionic models. (**a**) Layered (**b**) Columnar (**c**) Tubular (**d**) Helical (**e**) Sutured (**f**) Sandwich.

**Figure 14 biomimetics-07-00176-f014:**
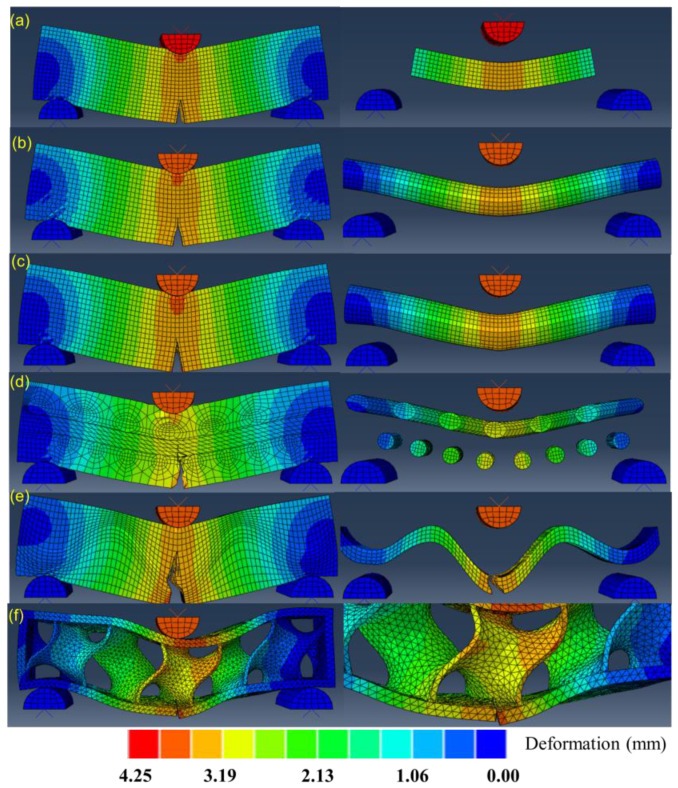
Simulations of crack propagations of the six bionic models (**a**) Layered (**b**) Columnar (**c**) Tubular (**d**) Helical (**e**) Sutured (**f**) Sandwich.

**Figure 15 biomimetics-07-00176-f015:**
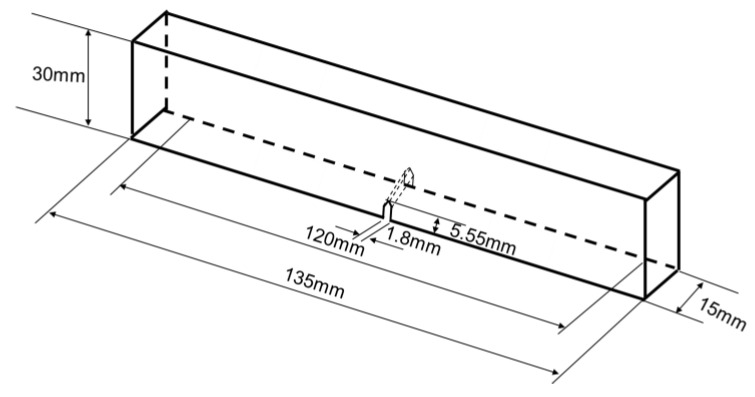
Sizes of the six specimens for three-point bending test.

**Figure 16 biomimetics-07-00176-f016:**
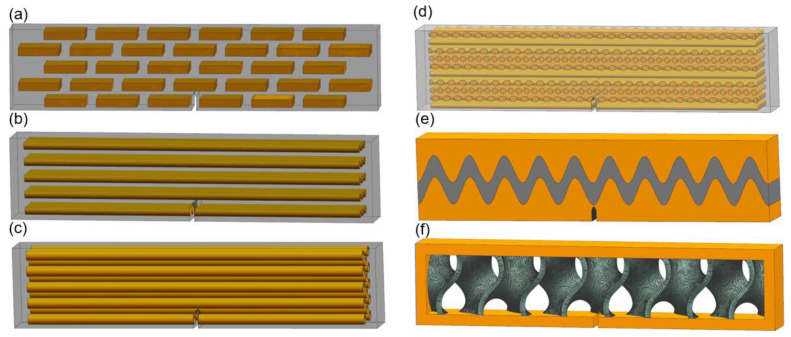
Sample design of the six biological models for three-point bending test. (**a**) Layered (**b**) Columnar (**c**) Tubular (**d**) Helical (**e**) Sutured (**f**) Sandwich.

**Figure 17 biomimetics-07-00176-f017:**
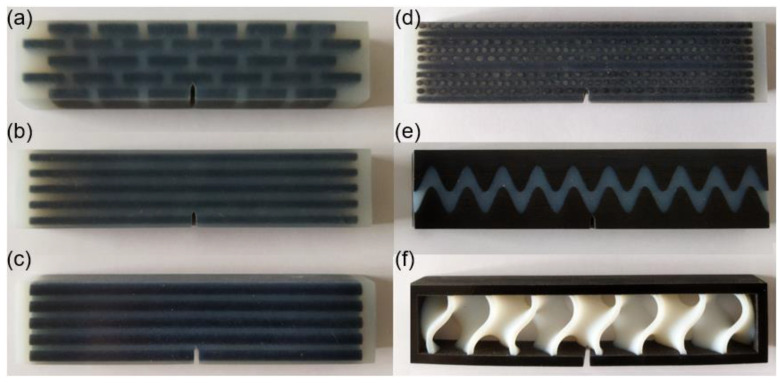
Test samples of the six bionic structures fabricated by 3D printing. (**a**) Layered (**b**) Columnar (**c**) Tubular (**d**) Helical (**e**) Sutured (**f**) Sandwich.

**Figure 18 biomimetics-07-00176-f018:**
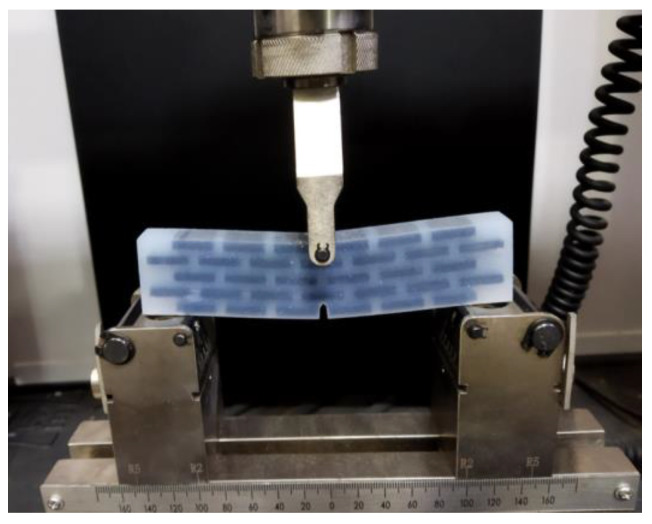
Three-point bending experiment of bionic layered structure.

**Figure 19 biomimetics-07-00176-f019:**
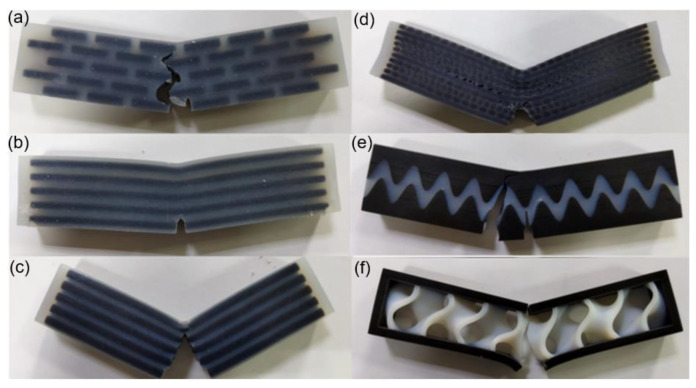
Experimental results of the six bionic structures. (**a**) Layered (**b**) Columnar (**c**) Tubular (**d**) Helical (**e**) Sutured (**f**) Sandwich.

**Figure 20 biomimetics-07-00176-f020:**
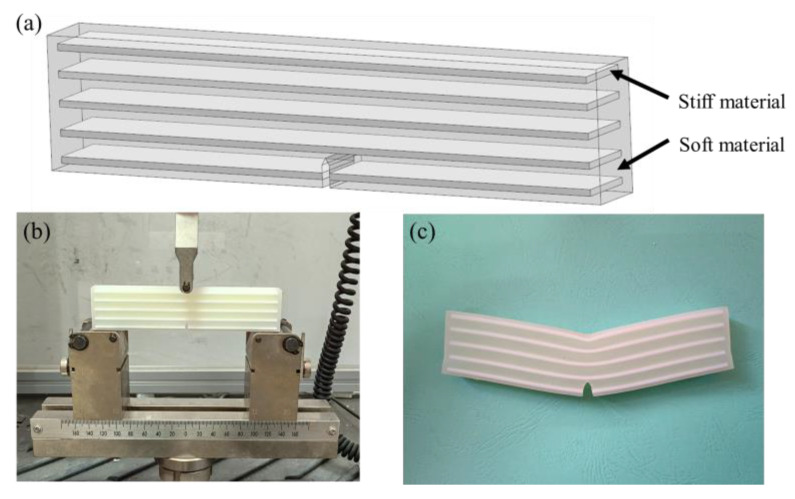
Experimental test of the reference sample. (**a**) Laminated model (**b**) Three-point bending experiment of conventional laminated structure (**c**) Experimental results of laminated structure.

**Figure 21 biomimetics-07-00176-f021:**
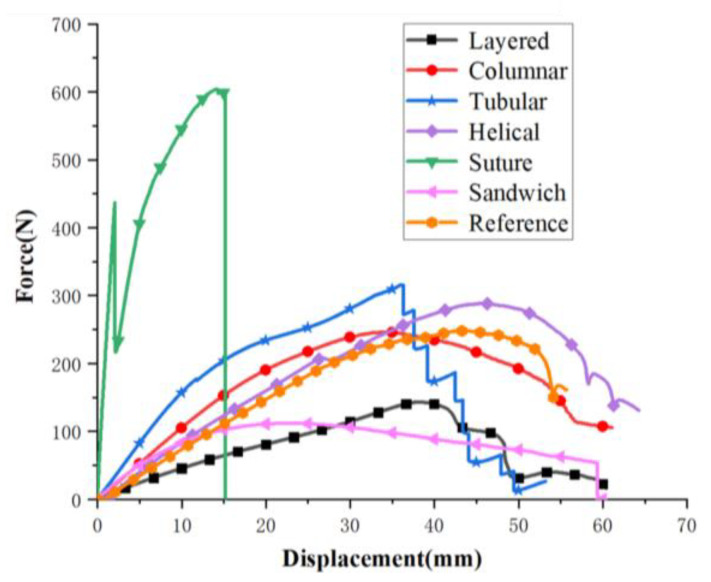
Analysis of experimental results of the six bionic structures and the reference model.

**Figure 22 biomimetics-07-00176-f022:**
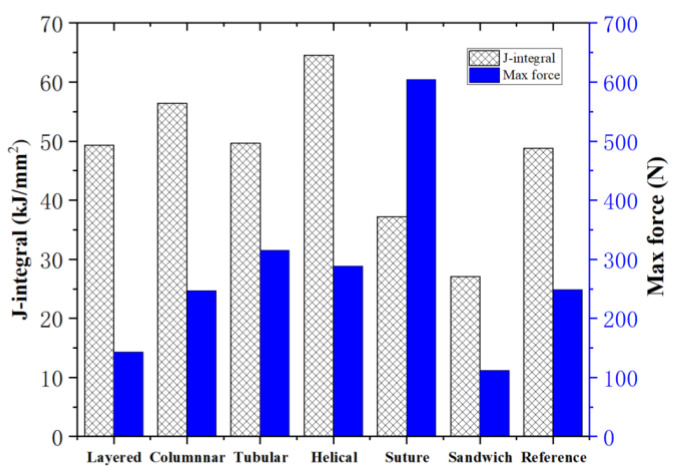
Maximum compression forces and toughness of the six bionic structures and the reference model.

**Table 1 biomimetics-07-00176-t001:** Materials parameters.

	Soft Matrix	Stiff Material
Young’s modulus (MPa)	600	3000
Tensile strength (MPa)	8.5	800
Elongation at failure	160%	10%
Density (g/cm^3^)	1.14	1.17
Poisson ratio	0.35	0.3
Shore hardness	50	86

**Table 2 biomimetics-07-00176-t002:** Materials failure parameters.

	Soft Matrix	Stiff Material
Max principal stress (MPa)	6.5	350–550
Displacement at failure(mm)	0.2	0.01
Viscosity coefficient	0.005	0.005

## Data Availability

The datasets generated during and/or analyzed during the current study are available from the corresponding author on reasonable request.

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
