# Peer review of "Bending Study of Six Biological Models for Design of High Strength and Tough Structures"

_biomimetics, 2022, doi:10.3390/biomimetics7040176_

Round 1
Reviewer 1 Report
29 - not clear what the authors mean
31 - not clear. what models are the authors referring to? numerical, design strategies, etc?
109 - can the authors provide more information on this model? what is the scale at which the model works? if we compare it with a fiber reinforced composite, each fiber could be considered as a column embedded in a matrix. However, strengthening of the interface has always been critical to enhance strength. so there must be an optimal scale at which the weakening of the interface provides optimal strength and toughness. Bear in mind that if the interface fails, the fiber cannot transfer load anymore. So, where is the balance?
136 - it is not clear how the tubular structure increases the strength of the material. It is clear how toughness can be increase via crack deflection. But why strength should increase with respect to a non-tubular structure? can the authors explain better?
166 - again - there is focus on toughness but not on strength. How is strength increased? and more importantly what type of strength? do helical structure improve all types of strength (e.g. compression, tension, flexural, shear, impact)?
275 – since this is a fracture mechanics problem, why not use classical fracture mechanics. Eg Griffith theory instead of a maximum stress principle? In such a way one can also introduce the quantity of material toughness in the problem.
302 – why the failure of the soft matrix is indicated by a certain applied displacement? This should be defined in terms of failure strain or the value of such displacement should be rather defined based on the fracture energy of the material rather than assigned explicitly.
306 – the model is attempted to also demonstrate crack deflection at the interface between soft and hard component. Now, such interface has usually specific properties which are quite different from the bulk properties of the soft matrix itself. Why not use a cohesive interface to demonstrate the capability of a certain structure to deflect a crack?
306 - the mesh seems to be very coarse around the notch tip. Did the authors performed a mesh sensitivity analyses to demonstrate that there is no mesh dependency?
316 - what results? there is no color-scale to know which variable has been shown in the figure. we can also notice excessive distortion of elements on the compressive side at the contact between the sample and the indenter (see Fig 14 d). Did the authors verified this?
All the figures should have improved captions with an explanation of what each subfigure represents.
370 – ASTM should have the number explicitly referenced
390 - the columnar still shows a softening, even though there is no crack propagation. What is this due to? the samples is in fact still failing. What about the fact that the different models have different hard component volume fraction? This might have a large effect on the opening capability of the initial notch.
408 - How does the J-integral value calculated with eq 2 compares against the J integral calculated via the FEM model? It is a simple operation to do in the FEM that could be easily confirm the applicability of Eq 2. Also the J integral is assuming a linear elastic behaviour along the contour and no edge effect. Can the authors justify this assumption? it seems that close to the indenter the fields are far from being elastic. Please, have a look at this paper for a more in depth explanation
D.M. Montenegro, L.P. Canal, J. Botsis, M. Zogg, A.R. Studart, K. Wegener,
On the validity of the J-integral as a measure of the transverse intralaminar fracture energy of glass fiber-reinforced polyurethanes with nonlinear material behavior,
International Journal of Solids and Structures,
Volumes 139–140,
2018,
Pages 15-28,
ISSN 0020-7683,
https://doi.org/10.1016/j.ijsolstr.2018.01.019.
Fig 21 – there several claims on the performance enhancement of the different microstructure but most likely the effect in different performance is also driven by the large change in volume fraction of the stiff component with respect to the soft component. Can the authors expand on this?
435-445 – it is not clear the purpose of citations 75 and 76. There seems to be no correlation to the current paper. Can the authors better explain this paragraph? If this is required to support the numerical model, this should go into the relative section and not in the discussion.
458 – while the biological models might have been at the micro and meso scale, the authors have all represented these models at the macro-scale. The hierarchical arrangement over multiple scales is a key feature of the outstanding performance of certain biological structures, therefore not representing each specific feature at the correct hierarchical level might make the comparison across the different models difficult.
The overall paper focus on achieving a good compromise between strength and toughness. However, during the whole discussion only notched samples and models have been investigated, therefore the “strength” reported by the authors is intrinsically related to the toughness of the model being everything approached as a fracture mechanics problem (there is always a notch in the microstructures investigated) and it is not a real representation of the actual unnotched strength of each microstructure. Furthermore, the reviewer feels that a reference model (not architected) is missing to be able to claim the actual improvement of nature-inspired design vs conventional designs. Additionally, I would be important to add a discussion on how the investigated model can improve current engineering practice and how such models can be reproduced at scale with conventional engineering materials.
Reviewer 2 Report
I've read with great interest the paper from Chen et al, entitled "Mechanism study of six biological models for high strength and tough structure and design".
Indeed, the biomimicry developments, in both civil and tissue engineering, are of tremendous importance for the future of the fields. The ideal combination of strength and toughness is a goal that living structures, through evolution, have been able to optimize.
In this study the authors highlight six biological models to be analyzed. Methodology is accurate and conclusions, in terms of biomechanics, are clearly supported.
I have comments for minor corrections:
- There should be an additional figure, with the 6 models represented at once: it is easier for the reader, and will provide a better overview.
- Although mentioned here, the aspect of "composite" structures is not investigated, i.e. columnar + tubular like in teeth: indeed, the final mechanical behavior is given by the whole, and not an addition of each composite layer. This should be discussed. And could represent hints for future investigations.
- Some mechanical properties are missing, like tensile, torsion etc... these are also new investigations that shall be conducted, in order to better illustrate the best model in each condition. Because, at the end for both civil and tissue engineering, the structural choice will rely on this type of complete investigations.
- Thus the current title: "mechanisms study" is not accurate and should mention it is bending study.
Round 2
Reviewer 1 Report
no further comments